


# **Natural marine bromoform emissions in the fully coupled ocean-**
# **atmosphere-model NorESM2**
Dennis Booge[1,2], Jerry F. Tjiputra[3], Dirk J. L. Olivié[4], Birgit Quack[1] and Kirstin Krüger[2]
[1]GEOMAR Helmholtz Centre for Ocean Research Kiel, Kiel, 24105, Germany
[2]Department of Geosciences, Section of Meteorology and Oceanography, University of Oslo, Oslo, 0371, Norway
[3]NORCE Norwegian Research Centre and Bjerknes Centre for Climate Research, Bergen, 5007, Norway
[4]Norwegian Meteorological Institute, Oslo, 0313, Norway
*Correspondence to*: Dennis Booge (dbooge@geomar.de) and Kirstin Krüger (kkrueger@geo.uio.no)
**Abstract.** Oceanic bromoform ($CHBr_3$) is an important precursor of atmospheric bromine. Although highly relevant for
the future halogen burden and ozone layer in the stratosphere, the global $CHBr_3$ production in the ocean and its emissions
are still poorly constrained in observations and are mostly neglected in climate models. Here, we newly implement marine
$CHBr_3$ in the state-of-the-art Norwegian Earth System Model (NorESM2) with fully coupled ocean-sea-ice-atmosphere
biogeochemistry interactions. Our results are validated with oceanic and atmospheric observations from the HalOcAt
(Halocarbons in the Ocean and Atmosphere) data base. The simulated mean oceanic concentrations (6.61±3.43 pmol L$^-$
$^1$) are in good agreement with observations in open ocean regions (5.02±4.50 pmol L$^{-1}$), while the mean atmospheric
mixing ratios (0.76±0.39 ppt) are lower than observed but within the range of uncertainty (1.45±1.11 ppt). The NorESM2
ocean emissions of $CHBr_3$ (214 Gg yr$^{-1}$) are in the range of or higher than previously published estimates from bottom-
up approaches but lower than estimates from top-down approaches. Annual mean emissions are mostly positive (sea-to-
air), driven by oceanic concentrations, sea surface temperature and wind speed, dependent on season and location. During
low-productivity winter seasons, model results imply some oceanic regions in high latitudes as sinks of atmospheric
$CHBr_3$, because of its elevated atmospheric mixing ratios. We further demonstrate that key drivers for the oceanic and
atmospheric $CHBr_3$ variability are spatially heterogeneous. In the tropical West Pacific, which is a hot spot for oceanic
bromine delivery to the stratosphere, wind speed is the main driver for $CHBr_3$ emissions on annual basis. In the North
Atlantic as well as in the Southern Ocean region the atmospheric and oceanic $CHBr_3$ variabilities are interacting during
most of the seasons except for the winter months where sea surface temperature is the main driver. Our study provides
improved process understanding of the biogeochemical cycling of $CHBr_3$ and more reliable natural emission estimates
especially on seasonal and spatial scales compared to previously published model estimates.
## 1 Introduction
Bromoform ($CHBr_3$) from the ocean is the most important organic compound for atmospheric bromine with an
atmospheric lifetime of ~2-4 weeks (Carpenter and Liss, 2000; Quack and Wallace, 2003; Salawitch, 2006; Papanastasiou
et al., 2014). As a reactive halogenated compound, it belongs to the very short-lived substances (VSLS) with lifetimes of
less than 6 months in the atmosphere (Law et al., 2007). In the tropics, VSLSs are rapidly lifted to the stratosphere by
tropical deep convection (Sala et al., 2014; Navarro et al., 2015; Fuhlbrügge et al., 2016), where they contribute up to
~25% to stratospheric bromine (Dorf et al., 2006 and following work). Bromine is ~60 times more efficient in depleting
lower stratospheric ozone than chlorine and significantly contributes to ozone depletion in the lower stratosphere (Daniel
et al., 1999; Sinnhuber et al., 2009; Montzka et al., 2011; Villamayor et al., 2023) with potential impacts on the radiation
budget of the atmosphere from -0.02 W m$^{-2}$ to -0.13 W m$^{-2}$ (Hossaini et al., 2015; Saiz-Lopez et al., 2023).
The oceanic air-sea gas exchange of $CHBr_3$ is parameterized based on the concentration gradient between surface water
and air and is related to wind speed and sea surface temperature via the transfer velocity (e.g. Nightingale et al., 2000).



Due to sparse measurements, marine $CHBr_3$ emission estimates are subject to large uncertainties (Laube et al., 2023).
$CHBr_3$ emission inventories from "bottom-up" approaches (e.g. Quack and Wallace, 2003; Butler et al., 2007; Ziska et
al., 2013; Lennartz et al., 2015; Stemmler et al., 2015; Fiehn et al., 2018) are based on in-situ oceanic data, whereas "top-
down" approaches (e.g. Warwick et al., 2006; Liang et al., 2010; Ordóñez et al., 2012) use in situ atmospheric mixing
ratio measurements. Resulting $CHBr_3$ emissions span a large range between 150 and 820 Gg Br $yr^{-1}$ (Laube et al., 2023).
The different methods cover e.g., statistical extrapolation of measurement-based data (Ziska et al., 2013; and update in
Fiehn et al., 2018), scaling of emissions to chlorophyll-a satellite observations (Ordóñez et al., 2012), modelling
atmospheric $CHBr_3$ with a modular flux in a chemistry climate model (Lennartz et al., 2015), and a data-oriented machine-
learning algorithm (Wang et al., 2019). These studies use limited spatial and temporal data coverage, underrepresenting
seasonal and interannual variations and spatial heterogeneity by averaging concentrations.
Oceanic $CHBr_3$ is mainly linked to primary production through natural processes such as marine organisms like
macroalgae and phytoplankton (Gschwend et al., 1985; Carpenter and Liss, 2000; Quack et al., 2004). Elevated surface
water concentrations are observed in coastal and shelf waters especially including the eastern boundary upwelling systems
(EBUS) (Quack and Wallace, 2003). Laboratory culture studies of phytoplankton production rates by Tokarczyk and
Moore (1994) and Moore et al. (1996) reported $CHBr_3$ increase during the exponential growth phase of phytoplankton.
Those specific growth rates and the corresponding temporal changes in $CHBr_3$ concentrations were first applied in a
physical biogeochemical water column model for the tropical Atlantic (Hense and Quack, 2009), and later implemented
in the global biogeochemical HAMburg Ocean Carbon Cycle model (HAMOCC; Stemmler et al., 2015). Stemmler et al.
(2015) explicitly implemented sources and sinks of marine $CHBr_3$ in the three-dimensional ocean biogeochemistry model
HAMOCC. However, they are not fully coupled with the atmosphere, and resulting emissions rely on fixed, prescribed,
extrapolated, observed atmospheric data of Ziska et al. (2013). Since the atmospheric concentrations are regulated by the
oceanic emissions, accurate estimates of atmospheric and oceanic $CHBr_3$ variability require such coupling, which can be
achieved using an Earth System Model (ESM).
Here, we present the first global model simulation of $CHBr_3$ in the fully coupled Norwegian ESM (NorESM2), where
$CHBr_3$ production is prognostically related to the primary production in the ocean taking natural biological processes into
account. We present results from a historical experiment focusing on the period 1990 to 2014 and compare them with
HalOcAt observations (https://halocat.geomar.de). Furthermore, we evaluate oceanic $CHBr_3$ excess and deficit regions
and use multilinear regression analysis to identify drivers of oceanic and atmospheric $CHBr_3$, as wells as $CHBr_3$ emission
variations on regional and temporal scales.
## 2 Model and Methods
We use the latest version of NorESM2 (NorESM2-LM; Seland et al., 2020; Tjiputra et al., 2020), which has participated
in the Coupled Model Intercomparison Project phase 6 (CMIP6) and contributed to the latest assessment report of the
IPCC-AR6 (Masson-Delmotte et al., 2023). The NorESM2 is a fully coupled ESM and is partly based on the Community
ESM Version 2 (Danabasoglu et al., 2020), which is developed by the National Center for Atmospheric Research (NCAR)
in the United States. NorESM2 is an updated version of its original version NorESM1 (Bentsen et al., 2013; Tjiputra et
al., 2013). It consists of a modified version of the Community Atmosphere Model version 6 (CAM6-Nor), the isopycnic
coordinate Bergen Layered Ocean Model (BLOM), the ocean biogeochemistry model isopycnic coordinate HAMOCC
(iHAMOCC), the sea ice model (Community Ice CodE version 5.1.2; CICE5.1.2), the Community Land Model version
5 (CLM5), and the river runoff model (MOdel for Scale Adaptive River Transport; MOSART). Both BLOM and
iHAMOCC apply a tripolar grid with a horizontal resolution of ~1° and 53 vertical isopycnic layers, while CAM6-Nor
and CLM5 share a common horizontal resolution of ~2° and 32 hybrid-pressure layers (lowest atmospheric layer



thickness: ~120 m) and a model top at 3.6 hPa (~40 km altitude). Here, we briefly highlight key features of iHAMOCC
as well as the $CHBr_3$ implementation (Section 2.1). The iHAMOCC ocean biogeochemical module is based on the original
work of Maier-Reimer (2012), has gone through several improvements and was later adapted to an isopycnic coordinate
ocean model (Assmann et al., 2010; Tjiputra et al., 2010). The model prognostically simulates inorganic carbon chemistry
following the standard Ocean Model Intercomparison Project (OMIP) protocol. It includes a Nutrient Phytoplankton
Zooplankton Detritus (NPZD) type ecosystem module, where the phytoplankton growth rate is constrained by multi-
nutrient limitation as well as ambient light and temperature. Particulate organic matters produced in the euphotic zone is
exported to the interior with a sinking velocity that increases linearly with depth before it is remineralized back to
inorganic carbon. The NorESM2 is able to simulate the observed large-scale pattern of surface primary productivity as
well as the regional seasonal cycle (Tjiputra et al., 2020).
**2.1 Bromoform module in NorESM2**
**2.1.1  Oceanic bromoform**
The marine $CHBr_3$ processes implemented in iHAMOCC comprise of advection (*adv*), production (*β*), air-sea gas
exchange (*F*), and three sink terms of: photolysis (*UV*), hydrolysis (*H*) and halogen substitution (*S*), as shown in Eq. (1).
The production and photolysis occur in the euphotic layer (top 100 m depth) of the model, whereas the air-sea gas
exchange is computed in the top-most layer of the ocean (upper 10 m). Advection and other sink terms are calculated
throughout the water column. The change over time of the oceanic $CHBr_3$ concentration is modelled as:

$$\frac{[CHBr_3]}{dt} = adv(CHBr_3) + \beta - F - UV - H - S. \qquad (1)$$


The parameterizations for the different processes are largely based on Stemmler et al. (2015). $CHBr_3$ is produced during
biological production as follows:

$$\beta = \beta_0 * (\frac{f_1 * Si(OH)_4}{K_{phy}^{Si(OH)_4} + Si(OH)_4} + \frac{f_2 * K_{phy}^{Si(OH)_4}}{K_{phy}^{Si(OH)_4} + Si(OH)_4}), \qquad (2)$$


where diatom and non-diatom contributing factors, $f_1$ and $f_2$, are set equally to 1. In contrast to Stemmler et al. (2015), the
bulk $CHBr_3$ production ratio ($\beta_0$) is modified and set to $2.4 \times 10^{-6}$ nmol $CHBr_3$ (mmol N)$^{-1}$, based on Kurihara et al. (2012)
and Roy (2010).
The air-sea gas exchange is calculated as follows:

$$F = k_w * (C_w - \frac{C_a}{H_{bromo}}), \qquad (3)$$


where $C_w$ and $C_a$ are $CHBr_3$ concentrations in the surface ocean and $CHBr_3$ mixing ratios in the atmosphere, respectively.
Positive emissions are defined as outgassing to the atmosphere. The temperature-dependent dimensionless Henry's law
solubility constant ($H_{bromo}$) is defined in Moore et al. (1995):

$$H_{bromo} = e^{13.16 - \frac{4973}{SST}}, \qquad (4)$$


with SST the sea-surface temperature in Kelvin. $k_w$ represents the gas transfer velocity calculated following Nightingale
et al. (2000) using the 10 m surface wind speed (*u*):

$$k_w = (0.222u^2 + 0.33u) * (\frac{660}{Sc_{bromo}})^{0.5}. \qquad (5)$$






The Schmidt number ($Sc_{bromo}$) for CHBr$_3$ is defined in Quack and Wallace (2003) using the sea surface temperature $SST$
in °C:

$$Sc_{bromo} = 4662.8 - 319.45 * SST + 9.9012 * SST^2 - 0.1159 * SST^3. \tag{6}$$


The loss term due to photolysis is computed as follows:

$$UV = I_{UV} * \frac{I_0}{I_{ref}} * e^{(-a_w * z)} * [\text{CHBr}_3], \tag{7}$$


where the decay time scale $(I_{UV})^{-1}$ is set to 30 days (Carpenter and Liss, 2000). $I_0$ and $I_{ref}$ are the prognostic incoming UV
radiation (i.e., 30% of shortwave radiation) and annual average irradiance at the surface layer, respectively. $z$ is the depth
and $a_w$ is the attenuation coefficient of UV radiation, set to 0.33 m$^{-1}$.
The loss term related to hydrolysis is estimated following Stemmler et al. (2015):

$$H = A_1 * e^{\left(-\frac{E_A}{RT}\right)} * [\text{OH}^-] * [\text{CHBr}_3], \tag{8}$$


with $A_1$, $E_A$, and $R$ set to 1.23x10$^{17}$ L mol$^{-1}$ min$^{-1}$, 107 300 J mol$^{-1}$, and 8.314 J K$^{-1}$ mol$^{-1}$, respectively (Washington, 1995).
$T$ is the seawater temperature in Kelvin.

Degradation due to halogen substitution (Eq. 5-6 of Stemmler et al., 2015):

$$S = L_{ref} * e^{\left(A_2 * \left(\frac{1}{T_{ref}} - \frac{1}{T}\right)\right)} * [\text{CHBr}_3], \tag{9}$$


with $L_{ref}$ and $A_2$ set to 7.33x10$^{-10}$ s$^{-1}$ and 12507.13 K, respectively, and $T_{ref} = 298$ K.
**2.1.2    Atmospheric CHBr$_3$**
CHBr$_3$ is implemented as a 3-dimensional tracer in the atmospheric model and is transported by the large-scale
atmospheric circulation and sub-grid scale processes (shallow and deep convection, and boundary layer turbulence). It is
removed in the atmosphere by photolysis:

$$\text{CHBr}_3 + h\nu \rightarrow 3\,Br, \tag{10}$$


and by reaction with the OH radical:

$$\text{CHBr}_3 + \text{OH} \rightarrow 3\,Br. \tag{11}$$


The reaction rate $k$ [cm$^3$ molecules$^{-1}$ s$^{-1}$] for the removal of CHBr$_3$ by OH in

$$\frac{d[\text{CHBr}_3]}{dt} = -k * [\text{CHBr}_3] * [\text{OH}] \tag{12}$$


is defined as follows:

$$k = 9.0 * 10^{-13} \exp\left(-\frac{360}{T}\right), \tag{13}$$






with $T$ the ambient temperature in Kelvin, and [CHBr$_3$] and [OH] in molecules cm$^{-3}$. The loss rate of CHBr$_3$ by photolysis
can be expressed by

$$\frac{d[\mathrm{CHBr_3}]}{dt} = -I\,[\mathrm{CHBr_3}], \tag{14}$$


where $I$ [s$^{-1}$] depends on the intensity of solar radiation and photo-physical properties of CHBr$_3$. The OH concentration is
a monthly-varying climatology obtained from a Whole Atmosphere Community Climate Model (WACCM) historical
simulation with full tropospheric and stratospheric chemistry (Gettelman et al., 2019).
CHBr$_3$ in the atmosphere has no other sinks than reaction with OH (annual mean CHBr$_3$ lifetime: ~46 days) and photolysis
(CHBr$_3$ lifetime: ~23 days) and is not affected by dry or wet deposition.

## 2.2 Model setup

A historical transient model run from 1850-2014, based on the CMIP6 protocol, was performed following a 500-year
preindustrial spin-up. The coupling of CHBr$_3$ between the ocean and the atmosphere is carried out with an hourly time
frequency exchanging the air-sea gas transfer. For analysis of the model climatology as well as for analysis of the model
validation with observations and further analysis of the driving CHBr$_3$ factors, daily model output data was averaged over
a period of 25 years (1990-2014) resulting in one mean value for each day of the year. The standard deviation of each
day reflects the variability within this time period. The 1990 to 2014 interval was chosen as most of the observations for
the model validation are within that time period, as compiled in the HalOcAt database (https://halocat.geomar.de, last
access: 13.10.2023).

## 2.3 Observations: HalOcAt database

The HalOcAt database, compiled by Ziska et al. (2013), updated by Fiehn et al. (2018) and by this study, is an observation-
based database for global oceanic and atmospheric data of short-lived halogenated compounds, such as CHBr$_3$. To date,
there are 9369 oceanic and 65179 atmospheric CHBr$_3$ measurements listed in 68 oceanic and 156 atmospheric datasets
(campaigns), respectively. The following criteria were applied to the observations in order to be used for model validation:
- Sampling locations with an ocean bottom depth less than 200 m or closer than 100 km to land were excluded.
- Sampling depth of oceanic CHBr$_3$ measurements had to be within the first 10 m of the water column in order to
be comparable to the CHBr$_3$ output of the upper surface ocean model layer (10 m depth).
- Maximum sampling height of atmospheric CHBr$_3$ measurements was set to 30 m altitude.
- Wherever applicable, individual measurements throughout one day were averaged to result in a daily averaged
surface ocean concentration or atmospheric mixing ratio in order to consider the same temporal resolution as the
daily model output. The coordinates of the respective averaged data points throughout a day were also equally
averaged. These locations were used to compare the observation with the closest grid point of the model output.

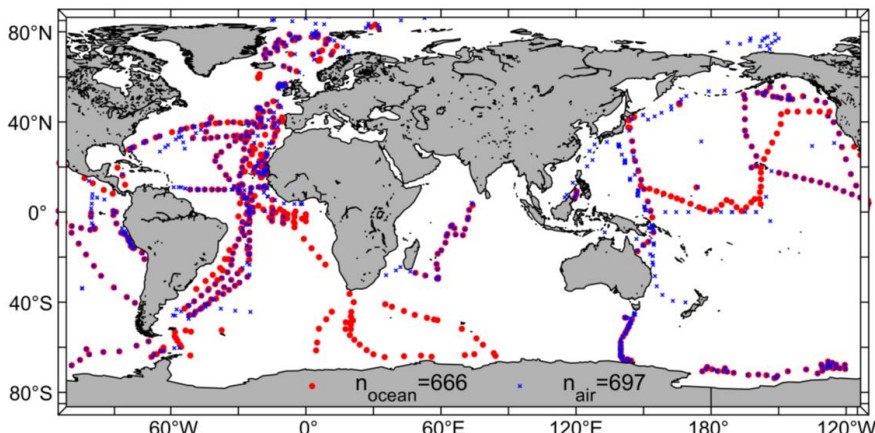

**Figure 1: Locations of oceanic (red, n=666) and atmospheric (blue, n=697) daily mean CHBr₃ observations from the HalOcAt database used to compare to daily mean NorESM2 model output.**

After screening the HalOcAt data base with the above-mentioned criteria, the individual oceanic and atmospheric datasets
(including the remaining datapoints) were tested for outliers. The mean from each dataset was calculated and the group
of all average values was tested for outliers. An outlier was defined as an element with more than three standard deviations
from the mean. According to the outlier test for oceanic and atmospheric datasets the corresponding dataset was removed
and not used for further validation of the model.
By addressing the mentioned criteria and datasets, we were able to validate the model with 666 daily mean oceanic (5154
individual) and 697 daily mean atmospheric (8411 individual) CHBr₃ observations from the HalOcAt database covering
both hemispheres (northern hemisphere (NH): 61%, southern hemisphere (SH): 39%), from the tropics (0-20°N/°S; 36%)
to the polar regions (60-90°N/S; 18%) with most observations in or above the Atlantic Ocean (44%) (Figure 1).
**2.4 Bromoform excess/deficit calculation**
The CHBr₃ excess/deficit (balance) rate ($k_{bal}$, Eq. (15), pmol m⁻² h⁻¹) is the difference between the CHBr₃ production rate
and the sum of different CHBr₃ loss rates, with all rates integrated over the upper 100 m depth):

$$k_{bal} = \sum production\ rate - \sum loss\ rate = k_\beta - (k_{UV} + k_F). \qquad (15)$$


The production term is described as the biological oceanic CHBr₃ production rate ($k_\beta$, Eq. (2)) and the loss term includes
the two fastest loss processes, i.e., photolysis due to UV radiation ($k_{UV}$, Eq. (7)) and the loss to the atmosphere via air-sea
gas exchange ($k_F$, Eq. (3)). We define a positive $k_{bal}$ as CHBr₃ excess rate and a negative $k_{bal}$ as CHBr₃ deficit rate. The
loss terms related to hydrolysis and to halogen substitution are not included as they are several orders of magnitude
smaller than $k_{UV}$ and $k_F$, in the surface ocean.
**2.5 Bromoform driving factor calculation**
Different parameters impact the variations of oceanic and atmospheric CHBr₃ values and influence the air-sea gas
exchange. These impacts can vary in magnitude and frequency dependent on local and seasonal conditions. Daily mean
average model output values from 1990-2014 were used to calculate annual and seasonally resolved (DJF, MAM, JJA,
SON) driving factors for oceanic CHBr₃ concentrations ($Bromo_{oce}$), atmospheric CHBr₃ mixing ratios ($Bromo_{air}$) and
CHBr₃ emissions ($Bromo_{flux}$) in three different specific areas (North Atlantic, tropical West Pacific, Southern Ocean),





which are presented in Section 3.4. Driving factors for each area, parameter and season were derived using multilinear
regression (MLR) analyses.
In order to compare parameters with different magnitudes, input data of each parameter was standardized prior to MLR
analysis by centering to have a mean of 0 and scaled to have a standard deviation of 1. Input data for each parameter
consisted of daily mean averages over the specific area, providing 365 values as basis for annually and ~ 90 values for
seasonally resolved MLR. Equations for MLR calculations were as follows with coefficients *a, b, c, d, e, f*, CHBr$_3$
production ($Bromo_{prod}$), $Bromo_{oce}$, $Bromo_{air}$ and $Bromo_{flux}$, as well as the 10 m surface wind speed (*wind*) and sea surface
temperature (*SST*):

$$Bromo_{oce} = a * SST + b * wind + c * Bromo_{prod} + d * Bromo_{flux} + e * Bromo_{air} + f \tag{16}$$


$$Bromo_{flux} = a * SST + b * wind + c * Bromo_{prod} + d * Bromo_{oce} + e * Bromo_{air} + f \tag{17}$$


$$Bromo_{air} = a * SST + b * wind + c * Bromo_{prod} + d * Bromo_{flux} + e * Bromo_{oce} + f \tag{18}$$


Other oceanic CHBr$_3$ loss processes (e.g. photolysis) were neglected in these calculations as the loss due to gas exchange
is ~70 times higher than the loss due to photolysis (data not shown). If the highest resulting coefficient for each season
and MLR is significantly higher than all other coefficients, the corresponding parameter is presented as the main driver
for either $Bromo_{oce}$, $Bromo_{air}$ or $Bromo_{flux}$. If the highest resulting coefficient is not significantly different from the second
or third highest coefficient, more than one coefficient and corresponding parameters are presented as main drivers. Table
1 lists the annual mean coefficients, Table S1 lists the seasonally resolved main drivers.

## 3  Results and Discussion

### 3.1  Model climatology

The annual and seasonal CHBr$_3$ oceanic concentrations, atmospheric mixing ratios and emissions reveal significant
spatial variations (Figure 2). The annual global average surface CHBr$_3$ concentrations are 5.04 pmol L$^{-1}$ (DJF:
5.36 pmol L$^{-1}$, JJA: 4.86 pmol L$^{-1}$) with highest annual mean concentrations of 28.37 pmol L$^{-1}$ in the upwelling region off
the coast of Peru and lowest annual mean concentrations of 1.37 pmol L$^{-1}$ in the Gulf of Boothia (71°N, 91°W) north of
Canada. The areas with the lowest oceanic CHBr$_3$ concentrations are the central parts of the North and the South Pacific
Gyres. Concentrations of surface ocean CHBr$_3$ in the entire NH (JJA: 5.9 pmol L$^{-1}$ > DJF: 4.3 pmol L$^{-1}$) and SH (DJF:
6.1 pmol L$^{-1}$ > JJA: 4.1 pmol L$^{-1}$) are generally higher during the respective summer than during the respective winter
season. These distinct differences of oceanic CHBr$_3$ concentrations are also due to the higher biological production in
summer (NH: 335 pmol m$^{-2}$ h$^{-1}$; SH: 371 pmol m$^{-2}$ h$^{-1}$) than in winter (NH: 235 pmol m$^{-2}$ h$^{-1}$; SH: 173 pmol m$^{-2}$ h$^{-1}$) as
shown in Fig. S3. The direct link of CHBr$_3$ to the biological production applies to the low oceanic CHBr$_3$ concentrations
in the North and South Pacific Gyres and to the high oceanic concentrations in the areas of the EBUS.
Variations in annual mean atmospheric CHBr$_3$ mixing ratios mainly follow the surface ocean concentrations with highest
mixing ratios in the tropics, especially in the EBUS. Global annual average mixing ratios over the ocean are 0.67 ppt
(DJF: 0.70 ppt, JJA: 0.69 ppt) with highest annual mean mixing ratios of 2.21 ppt in the south-eastern Pacific upwelling
region off the coast of Peru and lowest annual mean mixing ratios of 0.13 ppt over the Persian Gulf. On a global average,
the variability of atmospheric mixing ratios is lower than the variability of CHBr$_3$ concentrations in the surface ocean
(Figure 2). During austral winter (JJA), mostly dark and cold conditions increase the lifetime of atmospheric CHBr$_3$,
which leads to a uniform mixing ratio (0.67±0.05 ppt) over the entire Southern Ocean. Similar to oceanic CHBr$_3$



concentrations, central parts of the North and South Pacific Gyre have low atmospheric $CHBr_3$ mixing ratios
($0.46\pm0.05$ ppt). During austral summer (DJF) atmospheric mixing ratios increase further as strong biological activity
increases surface ocean concentrations, which enhance the oceanic emissions.
Generally, supersaturation of $CHBr_3$ in the world's ocean leads to emissions from the ocean to the atmosphere (defined
as positive). Global annual mean emissions are 268 pmol $m^{-2}$ $h^{-1}$ (DJF: 294 pmol $m^{-2}$ $h^{-1}$, JJA: 253 pmol $m^{-2}$ $h^{-1}$) with
highest annual mean emissions of 953 pmol $m^{-2}$ $h^{-1}$ in the upwelling region off the coast of Peru. In the tropical regions,
annual mean emissions of 427 pmol $m^{-2}$ $h^{-1}$ between 10°N and 10°S, add to atmospheric entrainment of oceanic $CHBr_3$
up into the stratosphere (Fiehn et al., 2018; Tegtmeier et al., 2020). Lowest annual mean emissions of -1 pmol $m^{-2}$ $h^{-1}$ are
modelled under ice free conditions in the Gulf of Boothia (71°N, 91°W) north of Canada (white regions in Figure 2) with
very low oceanic $CHBr_3$ production and low seawater temperatures. However, the atmospheric mixing ratios are
comparably high under these conditions. These conditions favour negative emissions, which, according to the results of
our fully coupled ESM, can be seen in the Arctic and Antarctic during winter season, confirming the results by Stemmler
et al. (2015) and Ziska et al. (2013) although with a lower amount.
Generally, the modelled $CHBr_3$ emissions are high, where the ocean concentration is high and the elevated emissions lead
to elevated atmospheric mixing ratios. However, due to oceanic transport processes, locations of high oceanic $CHBr_3$
emissions do not always coincide with locations of high oceanic $CHBr_3$ production (compare Figure 2 and Fig. S3). In
the northern part of the Bay of Bengal (>18°N) e.g., ocean concentrations during DJF are very high (average:
21.64 pmol $L^{-1}$), while the emissions are not as high compared to other ocean regions, due to low wind speeds. This leads
to a lower atmospheric mixing ratio than expected from the oceanic concentrations and shows that oceanic $CHBr_3$
concentrations and emissions as well as atmospheric mixing ratios show regionally different interdependencies, which is
addressed in detail in Section 3.4.

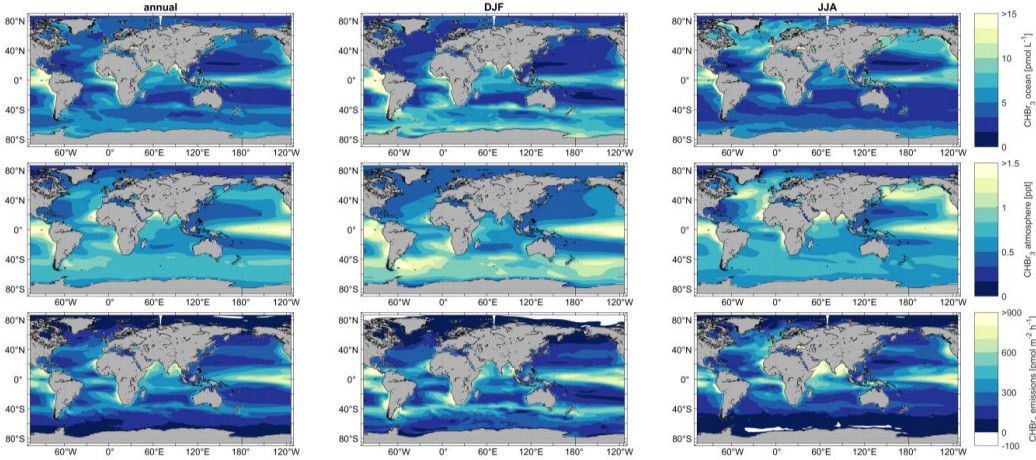

**Figure 2: Simulated annual (left), DJF (middle) and JJA (right) mean oceanic surface $CHBr_3$ concentrations (upper panel), atmospheric mixing ratios (middle panel) and $CHBr_3$ emissions (lower panel) for the period 1990-2014.**

**3.2 Model validation with observations**
The annual median surface oceanic $CHBr_3$ concentration (Figure 3a) from the 666 daily mean observations is 3.65 pmol $L^{-}$
$^1$ with a 25th and 75th percentile of 2.19 and 6.16 pmol $L^{-1}$, respectively (min: 0.05 pmol $L^{-1}$, max: 28.21 pmol $L^{-1}$, mean:
5.02 pmol $L^{-1}$). The global annual median surface oceanic $CHBr_3$ concentration from the model using only locations
corresponding with an existing observation is 6.00 pmol $L^{-1}$ with a 25th and 75th percentile of 4.23 and 8.10 pmol $L^{-1}$,



261 respectively (min: 1.39 pmol L$^{-1}$, max: 24.25 pmol L$^{-1}$, mean: 6.61 pmol L$^{-1}$). These results indicate that the model values

262 are in the range with observed concentrations of oceanic CHBr$_3$. While the median concentration of the model is higher

263 than the median of the observations, all validated model data points fall within the full range of the observations. The

264 model data cover a grid of ~100 km resolution, which leads to a smoothing of the values, whereas observational data is

265 local daily mean point data.

266 The median CHBr$_3$ atmospheric mixing ratio (Figure 3b) from the 697 daily mean observations is 1.17 ppt with a 25$^{th}$

267 and 75$^{th}$ percentile of 0.78 and 1.71 ppt, respectively (min: 0.03 ppt, max: 9.80 ppt). The global median atmospheric

268 mixing ratio of CHBr$_3$ from the model at locations with observations is 0.69 ppt with a 25$^{th}$ and 75$^{th}$ percentile of 0.49

269 and 0.90 ppt, respectively (min: 0.22 ppt, max: 2.70 ppt). This comparison shows that the observed atmospheric mixing

270 ratios of CHBr$_3$ are in the same magnitude but generally higher than those from the model output. While our model

271 experiment focuses on natural CHBr$_3$ production by phytoplankton, other sources as coastal macroalgae (Carpenter and

272 Liss, 2000) and anthropogenic sources, such as power plant cooling (Maas et al., 2021) or desalination plants (Agus et

273 al., 2009), may explain parts of the higher global annual median observational data of 41%. Jia et al. (2023) calculated

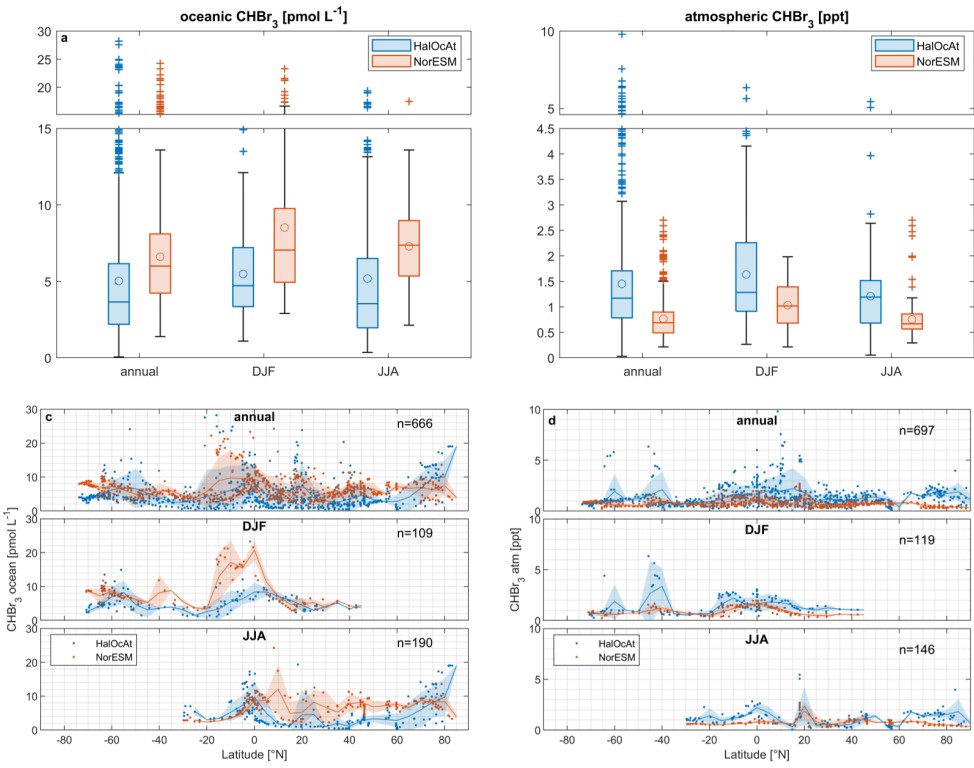

**Figure 3: Boxplot comparison of NorESM2 model results with HalOcAt observations for oceanic (a) and atmospheric CHBr$_3$ (b), c: zonal mean oceanic CHBr$_3$ comparison annually and in DJF and JJA, d: zonal mean atmospheric CHBr$_3$ comparison annually and in DJF and JJA. Shaded areas are standard deviations from 5° zonal bin averages. Boxplots (left) have a break in the y axis to increase readability of the figure. The line inside the box represents the median value, the circle the mean value, the boxes show the first to third quartile, and the whiskers illustrate the highest and lowest values that are not outliers. The plus signs represent outliers.**

274 an increase of global CHBr$_3$ emissions of 31.5% when including anthropogenic emissions, which partly explains also the

275 lower observed atmospheric mixing ratios in the model compared to the observations.

276 Figure 3 also shows a more detailed comparison between observations and model data in 5° meridional binned averages

277 (shaded areas) for oceanic (Figure 3c) and atmospheric (Figure 3d) CHBr$_3$ on annual basis as well as in JJA and DJF.

278 The modelled data compare well with observations of oceanic CHBr$_3$ (Figure 3c) on annual basis over the 5° latitudinal





bins. In the HalOcAt database, there are no oceanic and atmospheric observations available north of 50°N and south of
30°S during boreal (DJF) and austral winter (JJA), respectively, which highlights the need of model data to entirely
describe spatially and temporally resolved $CHBr_3$ (see also Fiehn et al., 2018). During DJF, the model overestimates the
measured concentrations between 20°N and 5°S. During JJA, averaged model concentrations in the NH (10°N – 60°N)
are slightly higher than the averaged observations. These discrepancies could indicate a missing process understanding,
revealing lower oceanic production or additional loss processes.
With all data available, the 5° latitudinal averaged atmospheric $CHBr_3$ observations have a large spread in the tropics
resulting in a high standard deviation (Figure 3d). The model results in this region are uniform with a much lower standard
deviation. During boreal winter (DJF) atmospheric $CHBr_3$ observations and model results show a good agreement, with
an exception at 40-50°S. In this latitude range, observational atmospheric $CHBr_3$ mixing ratios (>3 ppt, Figure 3d) were
recorded between 24° and 60° W in the South Atlantic in 2007 (Gebhardt, 2008). Gebhardt (2008) reports enhanced
biological production in the Argentinian shelf-break zone (55°-60° W) with elevated chlorophyll-$a$ concentration up to
4.5 µg L$^{-1}$. These values suggest also a high production of $CHBr_3$ and subsequent high emissions to the atmosphere.  The
prevailing westerly winds, transported the $CHBr_3$ enriched air masses eastward to the remote South Atlantic region in
2007, while in the model lower biological production entails lower atmospheric mixing ratios compared to the
observations. During boreal summer (JJA) very good agreement between atmospheric observations and model results is
obtained between 10°N and 60°N. North of 60°N, the model underestimates the measured atmospheric mixing ratios in
the polar region. Local meteorological and biological conditions (e.g. high wind speed, distinct phytoplankton blooms)
are averaged by the model to a resolution of ~100 km. Averaging data over time or space leads to lower values (e.g. gas
emissions, Bates and Merlivat, 2001), which explains lower modelled atmospheric mixing ratios compared to the
observations. These local and short term temporal variations contribute to the discrepancy in atmospheric values at global
scale as well as potential anthropogenic $CHBr_3$ emissions (Jia et al., 2023). Furthermore, discrepancies between model
results and observations also point to missing process understanding, which helps to improve our understanding of the
biogeochemical cycling of $CHBr_3$.
**3.3 Excess and deficit regions of oceanic bromoform**
In most of the world's surface oceans $CHBr_3$ production and loss rates are balanced on an annual average with a $k_{bal}$ close
to zero (e.g. North and South Pacific, top panel of Figure 4). The equator region experiences a strong excess rate (positive
$k_{bal}$) on annual average with values up to 300 pmol m$^{-2}$ h$^{-1}$ showing higher $CHBr_3$ production than loss of $CHBr_3$ in the
upper ocean, caused by strong primary production (Fig. S3) in the equatorial upwelling. Surface currents transport the
$CHBr_3$ enriched surface water masses away from the equator, while experiencing loss of $CHBr_3$ to the atmosphere.
Therefore, adjacent marine areas north and south of the equator experience a deficit rate (negative $k_{bal}$) of $CHBr_3$ (blue
areas, Figure 4), as no production balances the loss. The seasonality of $k_{bal}$ is pronounced in the extratropics (bottom
panels of Figure 4). In these regions, a $CHBr_3$ excess rate is observed mainly during summer and a $CHBr_3$ deficit rate
mainly during winter in the respective hemispheres. A high $k_\beta$ (elevated biological production) and a low $k_F$ (weak
emissions to the atmosphere) caused by lower winds during summer, lead to a higher $CHBr_3$ surface ocean concentration
in summer compared to winter time (Figure 2). During winter in both hemispheres, lower biological activity (low $k_\beta$) and
elevated wind speed (high $k_F$) decrease the $CHBr_3$ production and increase the emissions to the atmosphere, which leads
to a $CHBr_3$ deficit rate. These results reveal seasonal as well as spatial differences in parameters (driving factors), which
influence $CHBr_3$ concentrations in the world's ocean.





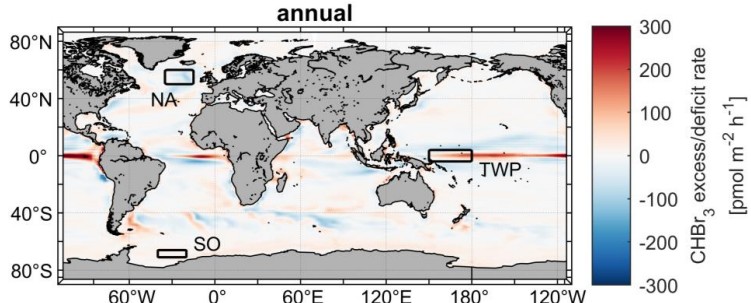

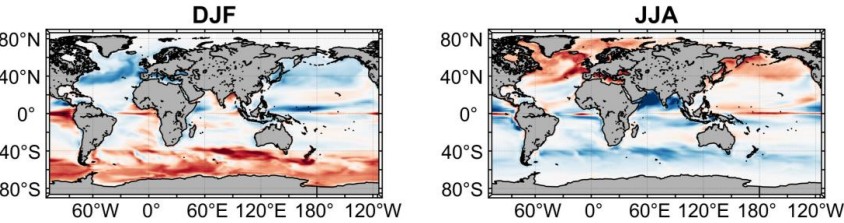

**Figure 4: Mean CHBr₃ excess/deficit rates on annual (top) and seasonal (DJF: bottom left; JJA: bottom right) basis. Three rectangles in the top figure illustrate locations of case studies. NA: North Atlantic; TWP: Tropical West Pacific; SO: Southern Ocean.**

In the following subsection, we selected three different case study areas, indicated in Figure 4, in order to contrast the
driving factors of the variations of oceanic and atmospheric CHBr₃ on regional and temporal scales:

320        -    North Atlantic, with an annual mean CHBr₃ deficit rate ($k_{bal}$ = -33 pmol m⁻² h⁻¹)

321        -    Tropical West Pacific, with an annual mean CHBr₃ excess rate ($k_{bal}$ = +32 pmol m⁻² h⁻¹)

322        -    Southern Ocean, with negative emissions during the respective winter season ($k_{bal}$ = +15 pmol m⁻² h⁻¹)

**3.4 Driving factors of bromoform on regional and temporal scales**
This section investigates the seasonal changes of oceanic and atmospheric CHBr₃ and other parameters in three
contrasting regions. Daily means of oceanic CHBr₃ concentrations, production, emissions, balance (as defined in Eq.
(15)), atmospheric mixing ratios as well as SST and wind speed, in the North Atlantic, tropical West Pacific and Southern
Ocean over an entire year reveal large differences between the regions (Figure 5). With MLR analysis, the main driving
factors of oceanic and atmospheric CHBr₃ variability and its emissions in each region and season are investigated.






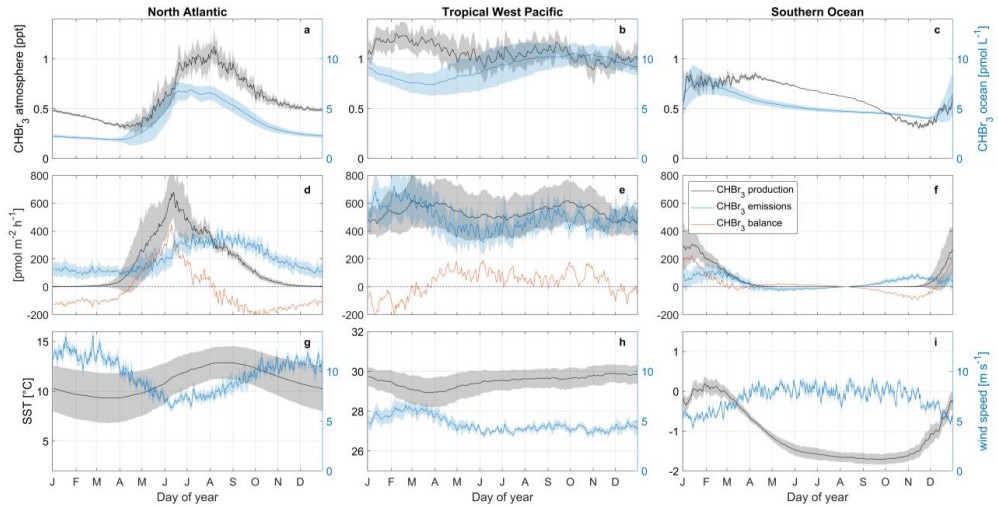

**Figure 5:** Seasonal changes of oceanic and atmospheric CHBr₃ (upper panel, a-c), CHBr₃ production, emissions and balance (middle panel, d-f), SST and wind speed (bottom panel, g-i), in the North Atlantic (left), tropical west Pacific (middle) and Southern Ocean (right). Shaded area is one standard deviation of the average value in the corresponding area. Note: y-limits for SST are not similar in between the three regions in order to increase readability of the figure.


**North Atlantic**
The North Atlantic region (50°N – 60°N, 15°W – 35°W) is characterized by a strong seasonal cycle of both oceanic
CHBr₃ concentrations and atmospheric mixing ratios (Figure 5a). The magnitude of the cycle is strongest among the three
investigated regions in this study (compare with Figure 5b,c). Oceanic CHBr₃ concentrations are on average 3.64 pmol L⁻
¹ with a minimum concentration of 1.87 pmol L⁻¹ during end of March and a maximum concentration of 6.93 pmol L⁻¹
during July. Atmospheric mixing ratios show a similar seasonal cycle, shifted by one month, with average values of
0.60 ppt, a minimum mixing ratio of 0.30 ppt during April and a maximum mixing ratio of 1.12 ppt during August. Figure
5d shows, that the CHBr₃ emissions (199±91 pmol m⁻² h⁻¹) follow the pattern of both oceanic and atmospheric values.
The seasonal cycle of CHBr₃ production (171±191 pmol m⁻² h⁻¹) is similar to the cycle of CHBr₃ concentration, while the
sharp peak in May/June when the spring phytoplankton bloom evolves in the North Atlantic, is not reflected in the oceanic
concentrations. The strong seasonality in CHBr₃ production leads to a CHBr₃ excess rate during summer (JJA:
103 pmol m⁻² h⁻¹) and a CHBr₃ deficit rate in winter time (DJF: -114 pmol m⁻² h⁻¹), respectively (Figure 5d).
The MLR analysis indicates, that on an annual basis, variations in atmospheric mixing ratios are mainly associated with
CHBr₃ ocean concentrations (Table 1, R²=0.89, p-value<0.05) and vice versa (Figure 6a,d). A higher surface water CHBr₃
concentration increases the emissions to the atmosphere resulting in increasing atmospheric mixing ratios. According to
the MLR analysis on a seasonal basis, oceanic CHBr₃ concentrations are mainly driven by the oceanic production during
MAM (R²=0.93, p-value<0.05) and SON (R²=0.99, p-value<0.05) (Table S1), which increases from March to June
sharply to 680 pmol m⁻² h⁻¹ before gradually decreasing in SON (Figure 5d). Annually, atmospheric mixing ratios are
mainly driven by the oceanic concentration (Figure 6d, Table 1, R²=0.89, p-value<0.05). This is also true on a seasonal
basis except for the winter (DJF) season, where SST is the main (indirect) factor influencing the atmospheric mixing ratio
variations (Table S1, R²=0.95, p-value<0.05), as lower SSTs increase the solubility of CHBr₃ and hardly any oceanic
CHBr₃ production occurs. Thus, the emissions of CHBr₃ decrease, even during comparably high wind speeds, which
leads to a decreasing atmospheric mixing ratio. CHBr₃ emissions are mainly driven by oceanic concentrations on an



annual basis (Figure 6g, Table 1, $R^2$=0.81, p-value<0.05). The MLR results further indicate that the driving factors are
highly variable, when looking on a seasonal basis. During spring (MAM), wind speed, SST, and $CHBr_3$ production are
almost equally driving the emissions (Table S1). During this season, $CHBr_3$ emissions are pretty constant at
130±29 pmol $m^{-2}$ $h^{-1}$ (Figure 5d). $CHBr_3$ production and SSTs slightly increase in spring, and an increase of the emissions
to the atmosphere is expected. However, emissions stay constant as surface wind speed decreases and lower the emissions.
This regional and seasonal pattern explicitly illustrates the interaction of different conditions influencing the $CHBr_3$
emissions. During summer (JJA), low winds and a high oceanic $CHBr_3$ concentration equally influence the increasing
emissions (Table S1). In contrast to spring, higher SSTs (lower solubility) are only of minor importance during JJA. In
autumn and winter, decreasing emissions are mainly driven by decreasing SSTs and, in DJF, additionally by high
atmospheric mixing ratios (in comparison to oceanic concentrations), which additionally dampens the emissions (Table
S1).

**Tropical West Pacific**

Figure 4 shows that the equatorial regions of the Atlantic and Pacific Oceans generally have positive $k_{bal}$ and therefore
are a source of oceanic $CHBr_3$, which is transported to other oceanic regions. $CHBr_3$ ocean concentrations in the tropical
West Pacific (4°S – 4°N, 150°E – 180°E) show a reduced seasonal cycle in comparison to the above discussed North
Atlantic region (Figure 5b). Oceanic concentrations are on average 9.11 pmol $L^{-1}$ which is significantly higher than
average concentrations in the North Atlantic while the seasonal amplitude (min: 7.42 pmol $L^{-1}$ in March; max:
10.58 pmol $L^{-1}$ in October) is less pronounced. $CHBr_3$ production (536±42 pmol $m^{-2}$ $h^{-1}$), $CHBr_3$ emissions
(492±84 pmol $m^{-2}$ $h^{-1}$) and atmospheric mixing ratios (1.07±0.08 ppt) show hardly any seasonality (Figure 5b, e). The
same is true for SST (29.50±0.28 °C) and wind speed (4.71±0.76 m $s^{-1}$) (Figure 5h). The $CHBr_3$ balance is positive
throughout the whole year except for DJF (Figure 5e). During this period high wind speed leads to higher emission than
production rates and induces low oceanic concentrations which results in a $CHBr_3$ deficit. However, this deficit does not
compensate for the $CHBr_3$ excess during the rest of the year leading to an overall positive $k_{bal}$ of 32 pmol $m^{-2}$ $h^{-1}$.
MLR analysis shows that the wind speed is the main factor influencing the variations of oceanic $CHBr_3$ concentrations
($R^2$=0.51, p-value<0.05), $CHBr_3$ atmospheric mixing ratios ($R^2$=0.74, p-value<0.05), and $CHBr_3$ emissions ($R^2$=0.73, p-
value<0.05) on an annual basis (Figure 6b,e,h, Table 1) in the equatorial region. During JJA and SON $CHBr_3$ production
drives the $CHBr_3$ concentrations (Table S1, JJA: $R^2$=0.43, p-value<0.05, SON: $R^2$=0.51, p-value<0.05) which increases
from 477 pmol $m^{-2}$ $h^{-1}$ in July to 618 pmol $m^{-2}$ $h^{-1}$ by the end of September. This results in an increase of oceanic $CHBr_3$
concentrations as all other parameters stay constant during that period.

**Southern Ocean**

The selected Southern Ocean region (71°S – 66°S, 40°W – 20°W) experiences water temperatures, which are negative
almost any time around the year (Figure 5i). Average temperature is -1.08°C with minimum temperatures of -1.71°C in
September (late winter) and maximum temperatures of +0.19°C in January/February (late summer). Wind speed in this
region is nearly constant throughout the year (7.33 m $s^{-1}$) with lower average wind speed of 5.76 m $s^{-1}$ only during austral
summer (DJF, Figure 5h). Oceanic $CHBr_3$ concentrations in the Southern Ocean region are on average higher
(5.38 pmol $L^{-1}$) than in the North Atlantic region with maximum concentrations (7.74 pmol $L^{-1}$) in January and lowest
concentrations (4.04 pmol $L^{-1}$) end of December. This sharp increase of $CHBr_3$ concentrations within two months
demonstrates the limited biological activity period, visible in the $CHBr_3$ production rate (Figure 5f). Due to decreasing
SSTs as well as a decreased day length, $CHBr_3$ production rates are almost zero from May to October and sharply increase
up to 306 pmol $m^{-2}$ $h^{-1}$ in January. Atmospheric mixing ratios are highest (max: 0.85 ppt) from January to beginning of
April but decline very slowly (Figure 5c) under low light levels, until they reach their minimum of 0.31 ppt in November.




Constant high atmospheric mixing ratios, due to light limitations in combination with very low SST, and decreasing
oceanic CHBr$_3$ concentrations, after the short summer bloom in DJF, influence the switch from positive emissions to
negative emissions between April and July (Figure 5f). CHBr$_3$ is in excess during times of CHBr$_3$ production (DJF) and
is almost balanced during the autumn and wintertime (April – September, Figure 5f). On annual basis, CHBr$_3$ is almost
balanced in this region with a slight excess of 15 pmol m$^{-2}$ h$^{-1}$.

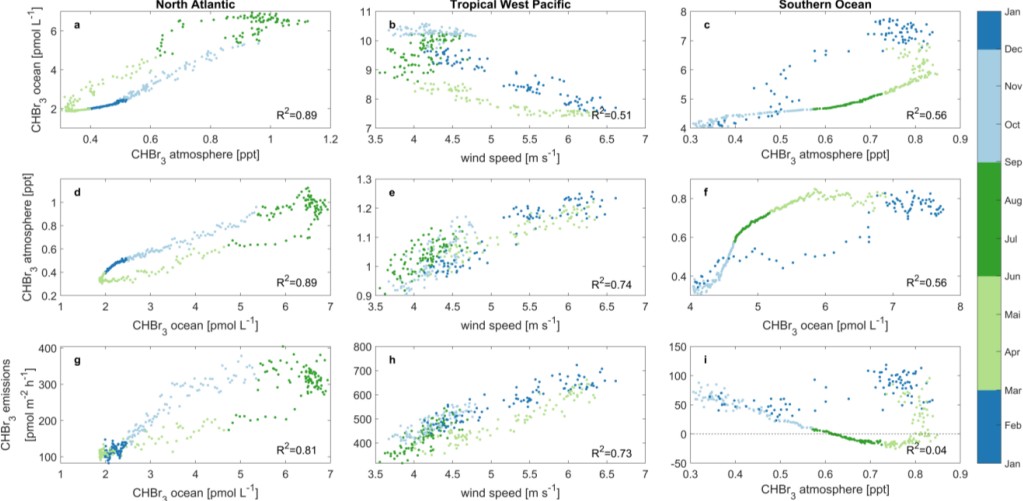

**Figure 6: Main drivers of oceanic CHBr$_3$ concentrations (a, b, c), atmospheric mixing ratios (d, e, f) and CHBr$_3$ emissions (g, h, i) in the North Atlantic (a, d, g), tropical West Pacific (b, e, h) and Southern Ocean (c, f, i). Different colours denote different seasons of the year. Each data point represents a daily mean average over the specific case study area.**

Overall, the MLR analysis confirms that atmospheric mixing ratios are the main factor influencing the variations of
oceanic concentrations on annual basis (Figure 6c, Table 1, $R^2$=0.56, p-value<0.05). CHBr$_3$ production is the driving
factor for the ocean concentration during autumn (MAM, Table S1). During this time CHBr$_3$ production decreases and
so does the ocean concentration Figure 5c,f). The atmospheric mixing ratio is mainly driven by oceanic concentrations at
times with high oceanic CHBr$_3$ concentrations (DJF: $R^2$=0.88, p-value<0.05) and by SST during cold winter times (JJA:
$R^2$=0.95, p-value<0.05) (Table S1). In winter during low light levels, atmospheric CHBr$_3$ reactions are reduced, which
increases the lifetime of atmospheric CHBr$_3$. Additionally, low SSTs increase the solubility of oceanic CHBr$_3$. These two
conditions favour the overall dampening of the CHBr$_3$ sea-to-air emissions during winter (JJA). During the summer in
DJF CHBr$_3$ emissions are mainly driven by SST (Table S1, $R^2$=0.60, p-value<0.05), as the solubility of CHBr$_3$ in the
ocean significantly decreases due to the increasing sea surface temperatures. After this short summer period, temperatures
decline in autumn (MAM) and increase the solubility of oceanic CHBr$_3$, which results in decreased emissions (Figure
5f,i). During winter (JJA) and spring (SON), surface temperatures and oceanic CHBr$_3$ concentrations stay low and
therefore, increasing emissions are mainly driven by decreasing atmospheric mixing ratios.

These results demonstrate the benefits of simulating CHBr$_3$ in a fully coupled ESM configuration to calculate driving
factors for different parameters on temporal and spatial basis. Studying the influence of one or more parameters on the
variability of other parameters in the model is not realistic when using prescribed oceanic concentrations or atmospheric
mixing ratios. Investigating the CHBr$_3$ cycling in different locations and different time scales helps to understand their
interconnection and to better integrate their results in today's as well as in a future climate.







**Table 1: Annual coefficients of predictors for each MLR in the different case studies. Bold coefficients are the highest value within a MLR analysis of one parameter and region and act as indicator for the driving factors of the predicted parameter (Eq. (16)-(18)).**

|  | Predictor parameter | North Atlantic | Tropical West Pacific | Southern Ocean |
|---|---|---|---|---|
| Ocean concentration | Wind speed | -0.02 | **-0.96** | -0.10 |
|  | SST | <0.01 | <0.01 | <0.01 |
|  | Atm. mixing ratio | **0.68** | 0.19 | **0.60** |
|  | $CHBr_3$ production | 0.39 | 0.13 | 0.53 |
|  | $CHBr_3$ emissions | <0.01 | <0.01 | <0.01 |
| Atmospheric mixing ratio | Wind speed | 0.29 | **0.94** | 0.55 |
|  | SST | 0.32 | <0.01 | <0.01 |
|  | Ocean concentration | **0.93** | 0.12 | **1.07** |
|  | $CHBr_3$ production | <0.01 | 0.02 | <0.01 |
|  | $CHBr_3$ emissions | <0.01 | <0.01 | <0.01 |
| $CHBr_3$ emissions | Wind speed | 0.20 | **1.27** | 0.21 |
|  | SST | 0.67 | 0.50 | 0.53 |
|  | Ocean concentration | **0.83** | 0.16 | 1.00 |
|  | $CHBr_3$ production | <0.01 | 0.10 | <0.01 |
|  | Atm. mixing ratio | -0.32 | -0.02 | **-1.22** |



### 3.5 Global bromoform emission inventories

A comparison of our modelled versus published global $CHBr_3$ emissions are presented in Figure 7. Global annual $CHBr_3$ emissions from top-down approaches are 449 Gg yr$^{-1}$, 528 Gg yr$^{-1}$ and 592 Gg yr$^{-1}$ based on calculations from Liang et al. (2010), Ordóñez et al. (2012) and Warwick et al. (2006), respectively. These inventories are about two to eight times higher than calculated annual emissions from bottom-up approaches, which are in the range of 76 Gg yr$^{-1}$ (Stemmler et al., 2015) to 238 Gg yr$^{-1}$ (Lennartz et al., 2015). Our results (214 Gg yr$^{-1}$) are similar to emission estimates published by Ziska et al. (2013) of 215 Gg yr$^{-1}$ but significantly higher than the 76 Gg yr$^{-1}$ estimate by Stemmler et al. (2015), which is based on the oceanic $CHBr_3$ observations from HalOcAt.

As we apply a 2.38 higher $CHBr_3$ production rate in the ocean as Stemmler et al. (2015), we simulate a production rate of 0.88 Gmol yr$^{-1}$ compared to 0.37 Gmol yr$^{-1}$ by Stemmler et al. (2015). Our emissions (214 Gg yr$^{-1}$) are 2.82 times higher (Figure 7, global values) compared to the emission estimate (76 Gg yr$^{-1}$) from Stemmler et al. (2015). Our model adaption is based on the higher bulk $CHBr_3$ production ratio ($\beta_0$) according to Kurihara et al. (2012) and Roy (2010) (see Section 2.1.1). This production rate is at the higher end of published values. Therefore, the resulting $CHBr_3$ production can be seen as an upper limit.

Comparing bottom-up and top-down approaches, the annual $CHBr_3$ emissions account for ~47% (105 Gg yr$^{-1}$) and ~66%
(351 Gg yr$^{-1}$), respectively, from the tropics (20°S – 20°N, Figure 7), which account for ~37% of global oceanic surface,
underlining the tropics as the most important source region of $CHBr_3$ of the earth.
Emissions in the middle latitudes (20 to 50°N/S) of the NH and SH show a similar distinction between top-down and
bottom-up approaches. However, the annual $CHBr_3$ emissions are only half that of the tropics. Natural open ocean
emission estimates from our study are proportional to the surface area between NH and SH in the middle latitudes. This
relationship is reversed for the top-down approach estimates. Top-down emission estimates are higher in the NH
compared to the SH although the oceanic surface area is lower in the NH (17%) compared to the SH (26%). This indicates
the strong influence of coastal emissions on observational atmospheric mixing ratios used in top-down approaches.
In the high latitudes (50-90°N/S), emissions of bottom-up approaches are in the same range (SH) and even higher (NH)

**Figure 7: Comparison of global and latitudinally binned annual $CHBr_3$ emissions from different studies. Grey and blue bars denote top-down and bottom-up approaches, respectively.**

compared to top-down approaches (Figure 7). In the northern polar region (8% of global oceanic surface area), $CHBr_3$ emissions from our study account for 3% (6 Gg yr$^{-1}$) of global emissions and are significantly lower than the other two bottom-up approaches from Lennartz et al. (2015) (11%, 27 Gg yr$^{-1}$) and Ziska et al. (2013) (21%, 45 Gg yr$^{-1}$), which appears mainly due to the resolved seasonality within our study. According to the HalOcAt database, no measurements are recorded from November to February and from May to September north of 50°N in the NH and south of 50°S in the SH, respectively. Therefore, the prescribed atmospheric values in Ziska et al. (2013) but also in Stemmler et al. (2015) are biased to the ice-free summer months, with higher atmospheric mixing ratios, thus artificially

in Stemmler et al. (2015) are biased to the ice-free summer months, with higher atmospheric mixing ratios, thus artificially
dampening the emissions from the ocean to the atmosphere during winter seasons. Due to the influence of the annually
fixed prescribed atmospheric mixing ratios in Stemmler et al. (2015), negative emissions are more pronounced between
50°N/S and 70°N/S up to -100 pmol m$^{-2}$ h$^{-1}$ at ~ 60°N/S. Our lesser negative emissions in the coupled ESM approach
appear more realistic as they are not based on summer biased prescribed values.

Our global $CHBr_3$ emission inventory indicates distinct differences to the top-down approaches reflecting only 40%-50%
of global emissions calculated by Liang et al. (2010), Ordóñez et al. (2012) and Warwick et al. (2006). Atmospheric
$CHBr_3$ values in the top-down approaches are higher than the calculated atmospheric mixing ratios from our fully coupled
model analysis. They include elevated coastal (Scenario A and C, Liang et al., 2010), including anthropogenic (Ordóñez
et al., 2012) sources, which may partly explain the discrepancy.
An additional explanation for the overall higher atmospheric mixing ratios of $CHBr_3$ from observations could be that
observations from coastal areas (100 km within the coastline) were excluded from this study and are not represented in
the model, as they are difficult to quantify (e.g. tide-dependent $CHBr_3$ emissions of macroalgae) with a horizontal model
resolution of 1°. However, coastal emissions lead to higher atmospheric mixing ratios of $CHBr_3$ (Fuhlbrügge et al., 2013;
Fuhlbrügge et al., 2016; Hepach et al., 2016), which can be transported to remote open ocean regions, while these higher
observational values are not included in the model results (Figure 3).



Another explanation for the underestimation of the modelled atmospheric mixing ratios compared to observations is the
use of air-sea gas exchange parameterizations, whose uncertainty is estimated to be 25% (Wanninkhof, 2007) and may
be underestimated up to 75% (Yang et al., 2022) at low wind speeds.

## 4    Conclusions and Outlook

Our study is the first one to derive oceanic and atmospheric $CHBr_3$ concentrations, as well as emissions, from a fully
coupled ESM simulation. The model prognostically simulates oceanic $CHBr_3$ production by phytoplankton and includes
oceanic $CHBr_3$ loss due to air-sea gas exchange, photolysis, hydrolysis and halogen substitution. Atmospheric loss of
$CHBr_3$ is described by photolysis and the reaction with OH. We validate the model results with more than 5,100 oceanic
and 8,400 atmospheric observations from the HalOcAt database. The simulated global mean $CHBr_3$ emission rate (214
Gg yr$^{-1}$) is in the range of previously published bottom-up approaches (76-238 Gg yr$^{-1}$), but significantly lower than top-
down approaches (449-592 Gg yr$^{-1}$). The model allows to realistically resolve seasonal and spatial variations and to
identify different drivers of oceanic and atmospheric $CHBr_3$ variability on regional and seasonal scales. Our results
indicate that only during high productive seasons a consequently high $CHBr_3$ production drives high oceanic $CHBr_3$
concentrations. During low productive seasons, relatively high atmospheric mixing ratios suppress the gas exchange and
consequently influence variations in oceanic $CHBr_3$ concentrations. In tropical regions (e.g. tropical West Pacific) with a
small seasonal cycle, but high oceanic concentrations and atmospheric mixing ratios, wind speed is the main factor driving
the variability of oceanic and atmospheric $CHBr_3$ and its emissions. The results clearly indicate the benefit of a fully
coupled ocean-atmosphere-biogeochemistry ESM. In earlier modelling studies, prescribed, fixed atmospheric or oceanic
values were applied, which bias the seasonal impact of different factors on oceanic and atmospheric $CHBr_3$ and
subsequently induce additional uncertainties to the magnitude of $CHBr_3$ emissions.
Our fully coupled ocean atmosphere approach resolves natural biogenic oceanic and atmospheric $CHBr_3$ including their
emissions at relatively high temporal and spatial model resolution. Validation with observational data shows good
agreement for large scale spatial patterns and we attribute the remaining model-data differences to missing coastal
sources, which are not implemented in the model. Comparison with other published $CHBr_3$ inventories indicates that
approaches without seasonality lack to resolve $CHBr_3$ emissions especially in high latitudes.
Our results demonstrate the potential for applying a fully coupled ESM to elucidate the primary drivers of the observed
$CHBr_3$ concentrations and emissions variability across spatial and temporal scales. Moreover, this model set-up allows
to implement additional oceanic derived VSLS in order to further investigate their influence on the atmospheric chemistry.
The dissociation of open ocean natural derived $CHBr_3$ from coastal area derived $CHBr_3$ in this study reveal that coastal
derived $CHBr_3$ influences open ocean atmospheric mixing ratios. Therefore, implementing natural coastal next to
anthropogenic sources and concurrent model resolution increase in these areas will help to further close the gap of
published $CHBr_3$ emission estimates between bottom-up and top-down approaches. Long-term future changes in $CHBr_3$
dynamics under various scenarios should be investigated with a fully coupled ESM, to study the impact of climate change
on $CHBr_3$ dynamics, e.g. in the Arctic, associated with loss of sea-ice and its climate feedback through interaction with
ozone chemistry.

**Data availability.** Observational data can be downloaded from https://halocat.geomar.de. Model data will be archived
and will be made available upon request.







**Author contributions.**

DB wrote the manuscript and led the discussion with contributions from all authors. DB analysed the model simulations
and prepared the graphics. JFT and DJLO implemented the CHBr$_3$ model code changes in NorESM2 in discussion with
BQ and all other authors. JFT carried out the model runs. KK led this project and initiated the research idea for this study.

**Competing interests.** The authors declare that they have no conflict of interest.


**Acknowledgements.** This work was financed by the Research Council of Norway through the KeyCLIM project

(295046) within the KLIMAFORSK/POLARFORSK program. Resources for the model simulations and data storage
were provided by Sigma2 - the National Infrastructure for High Performance Computing and Data Storage in Norway.

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
