# Peer review of "Natural marine bromoform emissions in the fully coupled oceanatmosphere-model NorESM2"

_Earth System Dynamics, 2024_

## Author Comment (AC1)

Response to referee #1:

Booge et al. presented a modeling study of bottom-up emission estimates for marine bromoform using a fully coupled ocean-atmosphere Norwegian Earth System Model, NorESM2. I think this is well done effort and it is very nice to see a comprehensive new bromoform emission estimate from a fully coupled ocean-atmosphere ESM. This study makes a great addition to the existing bromoform emission estimates, both bottom-up and top-down, and with further progress into the new era of a changing climate system. I support this paper to be accepted for publication in ESD, but I do have a few comments that should be addressed before the paper is published.

**We thank reviewer #1 for the positive evaluation of our work and for providing valuable comments in order to improve the manuscript. We are addressing the comments in the following (highlighted in bold). The lines refer to the originally uploaded manuscript.**

1. Page 7, L199-213. I think it would be more helpful to the readers if you can use a simple schematic diagram to illustrate these set of equations 16-18 that balance the oceanic, atmospheric concentrations, production, and flux. These terms are all inter-linked and the equations are practically identical, except the subscripts. The current way of trying to explain the relationships between these terms using just equations and text is not an optimal way, in my view.

   **We added a schematic diagram to the manuscript and moved Eqs (16-18) to the supplement. We now refer to this diagram when explaining the relationships between these terms in Section 2.5.**

[Figure]

Figure 1: Schematic illustration of relationships between different parameters influencing each other. Generic parameters in triangle influence the derived parameters in circles. Each derived parameter in a circle is influenced by all other five parameters. Relationships are the basis for the multilinear regression analysis using Eqs. (16-18).

2. Section 3.2, L257-273. It is very distracting to read through all these mean, $25^{th}$, $75^{th}$ percentiles, min and max. It also makes things harder when I want to compare the numerical values between observations and model output. I would suggest that you use $25^{th}$ & $75^{th}$ values as subscript & superscript for the mean values. If you really want to include the min and max, you can add them in the same way (sub & sup) with in parenthesis.

**We like this idea. To further increase the comparability between different numerical values we deleted the mean value as well as the $25^{th}$ and $75^{th}$ percentiles and added min. and max. values as subscript and superscript to the median value, respectively.**

3. Data availability and open data policy. I clicked on the link to https://halocat.geomar.de. It does not seem to me that these data are publicly available. The "click to join" link seemed to only let you submit an observation dataset, but nowhere on this page allows one to get access to data or even register to get an account to get access to data. This clearly does not meet open data policy that every journal is trying to abide by!

**Thanks for pointing out this issue. As some of the datasets within HalOcAt are not directly published yet, we added further information on the website how to receive the data for now. We changed the sentence in the "data availability" Section to: "Observational data can be made accessible by contacting the principal investigator of HalOcAt through https://halocat.geomar.de".**

4. L515-518. I fully agree. It would be very interesting to see if you can use NorESM2 in a future climate and see how winds, SSTs, and the ocean-atmosphere balance change CHBr3 emissions. I look forward to seeing future studies from the authors on this topic.

**Thanks a lot for the comment. We absolutely agree and are working on future climate scenarios of bromoform emissions.**

Minor comments:

1. Maybe it is more accurate to say annual mean fluxes, instead of emissions. When it is emission, it implies that it must be from ocean to atmosphere. Sinks is the corresponding term when flux values are negative, therefore from atmosphere to ocean.

**The reviewer is correct, that it is more accurate to say annual mean fluxes, instead of emissions. We changed "emissions" to "fluxes" when describing average values or describing fluxes in general. When specifically talking about fluxes from the atmosphere to the ocean we use the expression "negative fluxes". When talking about**

**fluxes from the ocean to the atmosphere we kept the expression "emissions". These terms are now defined in Section 2.1.1 (l. 109).**

2. Just say "winter", instead of "winter seasons". Short and adequate.

**Done.**

3. This is not a correct statement. The most important organic compound for atmospheric bromine is CH3Br, not CHBr3. But you can say it is "one of the most important …"

**Thanks for pointing this out. We changed the sentence to "Bromoform (CHBr3) from the ocean is one of the most important organic compounds for atmospheric bromine…"**

4. L33-34, you already said tropics at the beginning, you don't need to say "tropical" in the second half.

**We deleted the word "tropical" to just state "deep convection".**

5. L101-104, you need to describe what each term is in Eqn (2). I couldn't find descriptions of Si(OH)4 and KSi(OH)4phy.

**Thanks for pointing out this missing information. We added the following description referring to Eq (2): "…where $K_{phy}^{Si(OH)4}$ denotes the half-saturation constant for silicate (Si(OH)4) uptake…"**

6. I think you may be confused in terms of when to use "e.g.".The Latin abbreviation for "for example" is e.g., which stands for "exempli gratia.". For instance, L297-298 "Averaging data over time or space leads to lower values (e.g. gas emissions)" is not a correct way to use e.g. Gas emissions is not an example of lower values. L305, for example, (e.g. North and South Pacific) should be moved to after surface oceans

Besides, "e.g." should always be italic and with a "," after it. Make sure you look through all the e.g. in the text and fix when not appropriate.

**Thanks for this comment. We went through the manuscript and deleted "e.g.," where not appropriate. Whenever appropriate we added a "," after "e.g.". In L305, we moved "(e.g., North and South Pacific)" to after "surface oceans". According to the Copernicus style, "e.g." is not written in italic. Therefore, we kept it as is.**

1. L324-327. Move North Atlantic, tropical West Pacific, and Southern Ocean to the end of the first sentence.

   **Done.**

2. Change but also -> and

   **Done.**

3. I think you can simply say "winter emissions" here, instead of "the emissions from the ocean to the atmosphere during winter seasons". The current phrase is long and unnecessary. Emissions only occur from ocean to the atmosphere in this context.

   **We agree and changed it.**

---

## Author Comment (AC2)

Response to referee #2:

The manuscript by Booge et al. presents the first coupled ocean-atmosphere model in which bromoform is dynamically modelled in both the ocean and the atmosphere, opposite to previous studies in which prescribed concentrations in at least one compartment were used. The authors compare their modelled concentrations with available observations and assess environmental drivers for natural bromoform emissions from the ocean. They conclude that the remaining discrepancy between top-down and bottom up emission estimates likely result from coastal fluxes.

The paper is an important contribution to the field and will likely have a large impact by presenting the first fully-coupled dynamic model for natural bromoform emissions from the ocean. The study therefore fits the scope of the journal and I recommend it for publication, after some minor comments have been addressed.

**We thank reviewer #2 for reviewing this manuscript and providing helpful comments. We are addressing the comments in the following (highlighted in bold). The lines refer to the originally uploaded manuscript.**

Main comments:

Concerning the comparison between model and data: I think the authors could make more use of the potential of the model to guide further research on the marine cycling of bromoform by discussing remaining residuals. The discussion of errors ends with "discrepancies between model results and observations also point to missing process understanding, which helps to improve our understanding of the biogeochemical cycling of CHBr3." To which missing processes does the spatial and temporal distribution of remaining residuals point to? If modelled ocean concentrations are systematically too high everywhere in the ocean (Fig. 3a), are rather production rates too high or consumption rates too low (i.e. can the spatial and temporal distribution of residuals help to narrow this down)? I suggest to make the error and residual analysis more quantitative by using error metrics or 1:1 scatter plots and systematically discuss which processes may be missing, need an improved parameterization in the model or may need further experimental studies to describe rates and their dependencies.

**We thank the reviewer for his insight, and agree that several factors may lead to the discrepancies. To follow the reviewer's suggestion, we did a two-fold analysis of the residuals for oceanic and atmospheric results:**

**1.) We calculated the model bias (difference between model results and observations) for oceanic and atmospheric bromoform and added them as plotted below to the supplement (Figure S4). In general, bias is mostly positive in the ocean and mostly negative in the atmosphere. Moreover, there is no clear spatial pattern, which shows, that there is no spatial dependency of the bias, neither for oceanic nor for atmospheric bromoform.**

[Figure]

**Figure S4: Bromoform model bias of oceanic (upper panel) and atmospheric (lower panel) data used during this study. Red colours show a bias towards positive values (overestimation of modelled results compared to observations). Blue colours show a bias towards negative values (underestimation of modelled results compared to observations).**

**2.) We performed statistical tests if residuals show any spatial (i.e., latitudinal) or temporal (i.e., seasonal) dependencies. None of these tests turned out to be statistically significant. As an example, Figure R1 below shows the 1:1 scatter plot of modelled data and observations for oceanic and atmospheric bromoform including a plot of the respective residuals versus latitude. The scatter plots indicate that the model overestimates oceanic bromoform observations mostly at lower concentrations ($< 5$ pM), but underestimates the observed atmospheric concentration. This might be due to the high production rate and too weak ocean fluxes estimated in the model set-up and. The relatively high oceanic concentration can be seen as an upper limit of global bromoform production and is mentioned in Section 3.5 in the manuscript.**

[Figure]

**Figure R1: Upper panel: Scatter plots of observed vs. modelled oceanic (upper left) and atmospheric CHBr3 (upper right). Lower panel: residuals from scatter plots versus Latitude for oceanic (lower left) and atmospheric CHBr3 (lower right).**

**Although the statistical tests did not indicate further insights to missing processes we discuss some possibilities and added following text to the discussion at the end of Section 3.2:**

**"For our CHBr3 production rate, we used the highest production rate, which we could retrieve from the published data (Kurihara et al. (2012), Roy (2010)). Therefore, we likely do not underestimate the oceanic planktonic source in general, and either the production rates are too high or the sink rates are too low in some regions, e.g., the equatorial Pacific. Furthermore, the resulting model bias does not follow a spatial pattern (Fig. S4). We claim, that currently not enough observational or experimental information is available to narrow down on the answer. As pointed out, the underestimation of atmospheric CHBr3, despite the maximum of the planktonic CHBr3 source is likely due to averaging, a missing source for the atmosphere or even the parameterization of marine CHBr3 fluxes**

**yielding too low emission values. Despite the named uncertainties, which deserve further studies, the model reflects very well the data and therewith the current status of knowledge."**

Driving factors of bromoform on regional and temporal scales:

- This section is very long but remains rather descriptive and partly hard to read due to the data listed in the text. While shortening the descriptive part by transferring some data to a table to enhance readability, the discussion could be more streamlined and point to overall findings and implications from this analysis.

**Thanks for this comment. We have cleaned unnecessary or transferred some of the statistical data (i.e., R² and p-value) to Figure 7. Additionally, we added a paragraph summarizing the overall findings and implications from the three different regions:**

**"In summary, the three different regions clearly indicate that driving factors for atmospheric and oceanic bromoform as well as for bromoform fluxes are dependent on local conditions. Planktonic production, which is the only source for CHBr3 in the model set-up impacts the variability of oceanic CHBr3 concentrations only in regions with a distinct seasonality (i.e., North Atlantic, Southern Ocean) in biological production. During times of lower productivity, atmospheric mixing ratios influence the oceanic CHBr3 concentration. In subpolar and polar regions (i.e., Southern Ocean), oceanic CHBr3 and its subsequent fluxes are driven by its solubility related to the low SST in late winter and spring (i.e., sea-ice melt). Although wind speed is an important parameter for the air-sea gas flux, this study reveals that wind speed is only the main driver for oceanic and atmospheric CHBr3 variability in areas with low seasonality (i.e., Tropical West Pacific).".**

- Some parts of the section about the model climatology already discuss the driver of seasonal variation of CHBr3 concentrations, e.g. in relation to higher biological production (l. 223) or atmospheric mixing ratios (l. 235). Later on, biological production is not discussed in the section about drivers. It would make sense to bundle discussion about seasonality in one place.

**We agree with the reviewer that the influence for seasonal variations of bromoform concentrations are partly discussed in the section about the model climatology. However, these (seasonal dependent) links between bromoform concentrations and biological production are discussed in a more qualitative than quantitative way and for either global averages (i.e., NH vs SH in summer, l.221ff) or broad ocean regimes (i.e., Pacific Gyres). We also mention that these links are not always correct (i.e., Bay of Bengal, l.247ff) and refer to Section 3.4 for an in-depth analysis of driving factors in three specific regions. We agree that discussing "drivers" for seasonal variations in two different sections is confusing. Therefore, we avoided mentioning "driving factors" or similar in Section 3.1 rather stating that there are potentially links between a high bromoform production and high bromoform concentrations. Additionally, we are now referring in specific statements to Section 3.4 about these potential linkages (i.e., l.223, l.235) in order to clarify that this is the main section where statistically significant seasonal driving factors are presented and discussed.**

**We also would like to point out that our aim in Section 3.4 is to discuss the factors which drive oceanic and atmospheric bromoform as well as the fluxes. Driving factors for biological bromoform production are not discussed. This is simply due to the model constraints, as bromoform production is solely dependent on primary production (i.e., silicate, Eq. (2)).**

**However, biological bromoform production as a seasonal driver for oceanic bromoform variations is discussed several times (l.346, l.379, l.402) for the three different regions and respective coefficients are listed in Table S1.**

**As mentioned in the comment before we have updated Section 3.4. according to the reviewer's suggestion including a summarizing sentence about seasonal biological production as a driving factor for oceanic and atmospheric bromoform variability. We hope these overall changes in the section address also this concern raised by the reviewer.**

Global emission estimates:

- Is the higher emission estimate mainly the result of the larger production rate (which is 2.38 times larger, resulting in 2.82 times larger emissions than Stemmler et al., 2015? ), as written in l. 436ff? The discussion could be more specific here.

**Yes, the reviewer is partly right: The higher emission estimate between our study and Stemmler et al. (2015) is mainly based on the larger production rate. Moreover, the emission estimate ratio of 2.82 indicates an excess of 18% compared to the production ratio of 2.38. We hypothesize that this is also caused by the prescribed mean atmospheric values without any seasonality used in Stemmler et al. (2015). On annual average, these prescribed atmospheric values are too high (especially during winter) and artificially dampen the bromoform emissions from the ocean to the atmosphere. Therefore, our emission estimate is even higher by 18% compared to Stemmler et al. (2015) than the production ratio would indicate. We added this information to Section 3.5. to be more specific as the reviewer suggested:**

**"Moreover, the ratio of bromoform emissions (2.82) is 18% higher compared to the applied bromoform production ratio (2.38). This is caused by the prescribed mean atmospheric values without any seasonality used in Stemmler et al. (2015) and gets more significant in regions with pronounced seasonality of bromoform emissions such as at high latitudes. Especially during winter, the annually mean prescribed atmospheric values are too high and artificially dampen the bromoform emissions. This results in a higher emission estimate using our fully coupled model approach."**

Minor comments:

Title of section 2.5: suggestion "Calculation of drivers influencing bromoform concentrations and emissions"

**We thank the reviewer for the suggestion. "Bromoform concentrations" only refer to oceanic values as technically correct atmospheric values are "mixing ratios". Therefore,**

**we have slightly changed the reviewer's suggestion to "Calculation of drivers for oceanic and atmospheric CHBr3 and its emissions".**

l. 246: suggestion: "…although with a lower magnitude."

**Changed according to the suggestion of the reviewer.**

Fig. 5. Please change x-axis label, I assume it should be month of year, not day of year?

**Yes, it should be month of year. Changed.**

Fig. 6. Isn't panel a and d as well as c and f transporting the same information (just x and y axis flipped)? I assume that this is the case because ocean concentrations correlate to (i.e. "drive") atmospheric mixing ratios and vice versa, but it is a bit confusing to show the same data and relationship twice.

**We agree that it might be confusing, if the same data is shown (y and x-axis flipped) within one figure (a, d and c, f). This is the case, as the atmospheric bromoform drives oceanic bromoform and vice versa in the North Atlantic and Southern Ocean. We think by deleting 2 out of these 4 subplots might as well be confusing for the reader, as the stacked structure of the subplot is disturbed. Therefore, we added the following information to the figure caption, to not confuse the reader anymore but keeping the structure of the figure:**

**"Please be aware that subplots a and d as well as c and f transport the same information only with interchanged x and y axes, as both parameters, oceanic and atmospheric bromoform, are interdependent in the two regions."**

l. 442: something is missing in this sentence. Account for 44% of what?

**The reviewer is correct. Tropical emissions account for 47% of global bromoform emissions. We have rephrased the sentence:**

**"Comparing bottom-up and top-down approaches, the annual CHBr3 emissions from the tropics (20°S – 20°N, Figure 8) account for ~47% (105 Gg yr-1) and ~66% (351 Gg yr-1) of global emissions, respectively. The tropics only account for ~37% of global oceanic surface though, underlining this region as the most important source region of CHBr3 of the earth."**